# Learning to Merge Tokens via Decoupled Embedding for Efficient Vision Transformers

**Dong Hoon Lee**
KAIST
donghoonlee@kaist.ac.kr

**Seunghoon Hong**
KAIST
seunghoon.hong@kaist.ac.kr

## Abstract

Recent token reduction methods for Vision Transformers (ViTs) incorporate token merging, which measures the similarities between token embeddings and combines the most similar pairs. However, their merging policies are directly dependent on intermediate features in ViTs, which prevents exploiting features tailored for merging and requires end-to-end training to improve token merging. This paper proposes **D**ecoupled **T**oken **E**mbedding for **M**erging (**DTEM**) that enhances token merging through a decoupled embedding learned via a continuously relaxed token merging process. Our method introduces a lightweight embedding module decoupled from the ViT forward pass to extract dedicated features for token merging, addressing the restriction from using intermediate features. The continuously relaxed token merging, applied during training, enables us to learn the decoupled embeddings in a differentiable manner. Thanks to the decoupled structure, our method can be seamlessly integrated into existing ViT backbones and trained either modularly by learning only the decoupled embeddings or end-to-end by fine-tuning. We demonstrate the applicability of DTEM on various tasks, including classification, captioning, and segmentation, with consistent improvement in token merging. Especially in the ImageNet-1k classification, DTEM achieves a 37.2% reduction in FLOPs while maintaining a top-1 accuracy of 79.85% with DeiT-small. Code is available at `https://github.com/movinghoon/dtem`.

## 1 Introduction

Transformers [31] have become the dominant and most popular architecture in machine learning, excelling in various modalities and tasks. In computer vision, Vision Transformers (ViTs) [9, 30] have achieved state-of-the-art performance, outperforming conventional backbones in tasks such as image classification [30], object detection [4], and segmentation [15], as well as multimodal applications such as image captioning [33] and visual question answering [17]. A key factor in the success of ViTs is their ability to capture long-range dependencies between patches or tokens, regardless of their spatial positions, using self-attention. However, due to self-attention, ViTs have high computational and memory costs that increase quadratically with the number of tokens. Consequently, there has been significant interest in developing methods to improve the computational efficiency of ViTs.

In this pursuit, token reduction [26, 23, 39, 22] aims to progressively reduce the number of tokens in Transformer layers, often adhering to predefined reduction rates. Early approaches [26, 23, 39] propose to prune unimportant tokens based on their contribution to the task, as measured by scoring functions. Yet, simply pruning tokens leads to information loss, often resulting in significant performance degradation in high reduction rates. Alternatively, approaches based on token merging [21, 27, 40, 2, 20, 34, 12] aim to combine redundant tokens instead of removing them. Such redundancy is measured by the similarity between the tokens based on intermediate features in ViT, such as token- or key-embeddings. Token merging has several advantages over pruning; it can

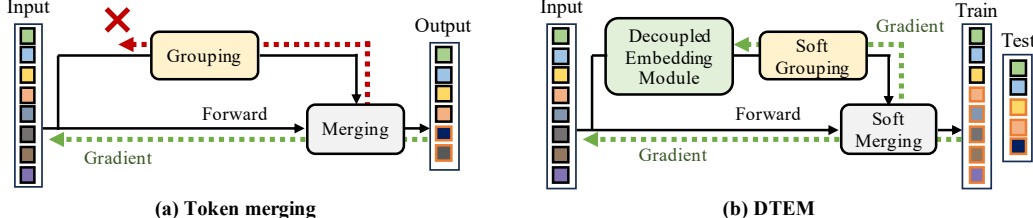

Figure 1: **Comparison of our method with conventional token merging.** Contrary to prior works that merge tokens directly based on intermediate features in ViT, our method leverages a decoupled embedding to extract features tailored for token merging. The embedding module is trained via continuous relaxation of grouping and merging operators, *i.e.*, soft grouping and merging, respectively, that allow differentiation.

achieve improved performance by reducing information loss in token reduction and can be seamlessly plugged into pre-trained models without altering the architecture.

However, merging tokens directly based on intermediate features, which are responsible for contextual encoding, presents several limitations. Firstly, these features are hard to be tailored specifically for token merging. This is because the same intermediate feature should be used for contextual encoding and merging; thereby, it would be less effective than having separate features dedicated to each role. Secondly, enhancing the merging process, which entirely relies on intermediate features, necessitates end-to-end training of the entire network. This makes it difficult to leverage pre-trained models effectively and typically requires extensive data to prevent overfitting.

To this end, we propose Decoupled Token Embedding for Merging (DTEM) that learns decoupled token embedding specifically tailored to enhance token merging. We introduce a lightweight trainable embedding module decoupled from the intermediate feature in the ViT and use it to modulate the merging policy. This resolves the dependency of token merging on intermediate features and facilitates the decoupled embedding to extract only suitable features for enhanced token merging. Moreover, since the modules are separated from ViTs, improved merging can be achieved without altering the ViT parameters, allowing for efficient modular optimization with pre-trained models.

However, learning the decoupled embedding module directly from conventional token merging is infeasible, since the grouping policy, *i.e.*, deciding which tokens to merge, is based on discrete operators such as hard cluster assignment [21, 40] or matching on a bipartite graph [2] (Figure 1(a)). To address this, we design a continuous relaxation of token merging that softly merges tokens in a differentiable way according to their similarities (Figure 1(b)). The relaxed operators, applied during training, enable training of the decoupled embedding directly through the grouping policy to improve token merging. We also observe that such relaxed operators tend to facilitate generalization of the learned decoupled embedding across unseen token reduction rates. During inference, our model converges to existing token merging methods by replacing the relaxed operators with hard ones.

We integrate DTEM in two distinct ways: modular and end-to-end full fine-tuning. For the former, we train only the embedding module while keeping the parameters of the pre-trained model frozen, while later we train the entire parameters in an end-to-end manner. We apply DTEM to existing pre-trained vision models and verify its effectiveness in image classification, captioning, and segmentation, each requiring a different level of granularity in representation. Despite the simplicity, DTEM consistently improves token merging in all three tasks, offering a better trade-off between task performance and computation cost. We further analyze DTEM's components, design choices, and training efficiency. Overall, our contributions are summarized as follows:

- We propose DTEM, a novel approach to enhance token merging by decoupled token embedding learned via continuous relaxation of token merging. The decoupled embedding is dedicated to merging and learns features suitable for merging directly from our relaxed token merging.

- DTEM can be applied through end-to-end full fine-tuning or in a modular way by training only the added embedding module. When trained modularly, the method delivers improvements even with substantially smaller datasets and fewer training epochs.

- Empirical evaluations over image classification, captioning, and segmentation across various ViT models demonstrate that DTEM consistently outperforms the prior arts.

## 2 Background

Given a Transformer that takes $N$ input tokens $\boldsymbol{X} \in \mathbb{R}^{N \times d}$, the objective of token merging is to gradually *merge* $r$ tokens at each Transformer block, reducing the number of tokens to $\hat{\boldsymbol{X}} \in \mathbb{R}^{(N-r) \times d}$. Here, the $r$ denotes the reduction rate. To this end, prior works conduct the merging in two steps, *grouping* and *merging*, which are expressed as:

$$\boldsymbol{E} = \text{Group}(\boldsymbol{S}), \tag{1}$$

$$\hat{\boldsymbol{X}} = \text{Merge}(\boldsymbol{X}, \boldsymbol{E}), \tag{2}$$

where $\boldsymbol{S} \in \mathbb{R}^{N \times N}$ denotes the similarity matrix of tokens, *e.g.*, $s_{ij} = \cos(\boldsymbol{x}_j, \boldsymbol{x}_j)$. Given the similarity $\boldsymbol{S}$, the grouping operator (Eq. 1) identifies pairs of tokens to merge and represents them in the reachability matrix $\boldsymbol{E} \in \{0, 1\}^{N \times N}$ with $(N - r)$ connected components, where $e_{ij} = 1$ indicates that the $i$th and $j$th tokens belong to the same component and will be merged. The merging operator (Eq. 2) then combines all connected tokens in $\boldsymbol{E}$ by pooling.

The performance of the above framework highly depends on the choice of the grouping operator, as it dictates the merging policy (*i.e.*, which tokens to merge), and computing the reachability matrix can be costly. Early works employ clustering algorithms [21, 40], but they tend to be slow due to the iterative procedure and often suffer from performance drops due to the dramatic distribution shift from $\boldsymbol{X}$ to $\hat{\boldsymbol{X}}$ caused by aggressive clustering.

Recently, ToMe [2] introduced Bipartite Soft Matching (BSM) as an efficient grouping operator of Eq. 1. To parallelize the computation, BSM divides the input tokens into two disjoint sets $\mathbb{A}$ and $\mathbb{B}$, and constructs a bipartite graph. Then for each node $i \in \mathbb{A}$, it chooses an edge with highest similarity $\arg \max_{j \in \mathbb{B}} s_{ij}$, and choose the $r$ most similar edges afterwards to obtain the sparse adjacency matrix $\boldsymbol{E}' \in \{0, 1\}^{|\mathbb{A}| \times |\mathbb{B}|}$ where $\sum_{ij} e'_{ij} = r$. The merging is performed by combining the connected tokens in $\boldsymbol{E}'$, where the connected components can be easily found since each token in $\mathbb{A}$ has at most one edge. ToMe [2] also proposes tracking the size of the combined tokens and accounts for it in self-attention. Specifically, given a vector $\boldsymbol{m} \in \mathbb{R}^{N-r}$ representing the size of combined tokens, the *proportional attention* is used in the QKV self-attention layers by:

$$\boldsymbol{A} = \sigma\left(\frac{\boldsymbol{Q}\boldsymbol{K}^\top}{\sqrt{d}} + \log \boldsymbol{m}\right), \tag{3}$$

where $\sigma$ denotes the softmax function.

**Limitations** Although the success of merging depends mostly on grouping operation (Eq. 1), the grouping depends entirely on the similarity of the intermediate ViT feature ($\boldsymbol{X}$ or $\boldsymbol{K}$) in the prior works. This is mainly because the grouping comprises discrete operators, such as clustering and matching, that prevent the gradient flow through grouping (Eq. 1). Thus, the only viable option to improve the merging is by updating the intermediate feature $\boldsymbol{X}$ by back-propagating through the merging operator (Eq. 2). However, it leads to extensive end-to-end training of the entire network, preventing off-the-shelf usage and resulting in suboptimal performance due to the conflict between the token feature required for optimal merging and task performance.

We provide more discussion on related work in the supplementary materials (A.3).

## 3 Method

Our objective is to improve token merging by learning the decoupled embedding specifically tailored for merging. To this end, we base our method on the standard token merging framework introduced in the previous section (Eqs. 1, 2). Instead of directly leveraging the ViT features $\mathbf{X}$ for grouping, we propose to learn additional per-token embedding modules $\boldsymbol{Z} = f(\boldsymbol{X}; \phi)$, which are decoupled from the forward pass of the ViT and used only to compute the similarity $\boldsymbol{S}$ in Eq. 1 by $s_{ij} = \cos(\mathbf{z}_i, \mathbf{z}_j)$ (Sec. 3.1). Since the grouping operator is entirely dependent on similarity, we can directly modulate the grouping (or merging) policy by learning $\mathbf{Z}$. Furthermore, since the embedding is decoupled from the ViT forward pass, enhancements in merging can be achieved modularly without altering the ViT parameters but only learning the embedding $\mathbf{Z}$.

To enable our model to learn such embeddings through merging, we propose a continuous relaxation of the grouping and merging operators in Eq. 1 and Eq. 2, respectively. Specifically, our relaxed

grouping operator generates a continuous matrix $\tilde{E}$, whose elements $\tilde{e}_{ij} \in [0, 1]$ indicates the *soft* degree of merging $i$th token into the $j$th token (Sec. 3.2). To incorporate such soft commitment in merging, we also propose a relaxed merging operator that combines tokens with continuous weights defined by $\tilde{E}$ (Sec. 3.3). Since the token merging is performed continuously with our relaxed operators, we discretize them after the training, reducing our framework to behave similarly to the hard merging methods [2] (Sec. 3.4).

## 3.1 Decoupled Embedding Module

We first describe the choice of the embedding module decoupled from the forward pass of ViTs. To facilitate token merging at each Transformer block, we introduce per-token projection layers into each block $l \in \{1, \dots, L\}$:

$$\text{Decoupled embedding } \boldsymbol{Z} : \boldsymbol{z}_i = f(\boldsymbol{x}_i; \phi_l), \tag{4}$$

$$\text{Token similarity } \boldsymbol{S} : s_{ij} = \cos(\boldsymbol{z}_i, \boldsymbol{z}_j), \tag{5}$$

where $\boldsymbol{X} \in \mathbb{R}^{N \times d}$ is the input to the self-attention and $\boldsymbol{Z} \in \mathbb{R}^{N \times d'}$ denotes the output decoupled embedding with $d' \ll d$. The output embeddings will be used *solely* to shape the merging policy (*i.e.*, deciding which tokens to merge) in the grouping operator based on the similarity $\boldsymbol{S}$.

Minimizing additional run-time and computational overheads is essential to the embedding module design. In our approach, we employ a token merging between the self-attention and feed-forward layer following [18, 2]. It allows parallelizing the computation of the attention and decoupled embedding, avoiding the potential overhead that comes from serialization. Moreover, we discover that even a shallow module, consisting solely of an affine transformation, can achieve improvement with minimal computational expense (Sec. 4.4). This further minimizes the number of additional parameters to less than 1% and enables the training of the module with a small amount of data.

## 3.2 Soft Grouping

Given the similarity matrix $\boldsymbol{S}$ obtained from the decoupled token embeddings $\boldsymbol{Z}$, soft grouping aims to approximate the grouping operation through a continuous relaxation that enables differentiation. However, building a general continuous grouping operator of Eq. 1 is challenging since the output reachability matrix is inherently discrete.

Instead, we employ BSM [2] as our target grouping operator, which offers the benefit of bypassing the reachability matrix and allows for merging to be defined directly on the adjacency matrix $\boldsymbol{E}' \in \{0, 1\}^{|\mathbb{A}| \times |\mathbb{B}|}$. To be a valid approximation of the grouping performed by BSM, the soft grouping operator should produce a continuous adjacency matrix $\tilde{\boldsymbol{E}} \in [0, 1]^{|\mathbb{A}| \times |\mathbb{B}|}$ that satisfies two key conditions. Firstly, it should simulate $r$ distinct edges with high values, thereby implementing the valid merging policy, *i.e.*, combining the $r$ most similar token pairs. Secondly, each node in $\mathbb{A}$ should be associated with at most one edge (*i.e.*, $\sum_{j \in \mathbb{B}} \tilde{e}_{ij} \leq 1$) to simplify the identification of connected components in $\tilde{\boldsymbol{E}}$, thus avoiding the complexity of computing a reachability matrix.

To achieve this, we propose a soft grouping that revises the differentiable top-$k$ operator from [25]. Starting with $\boldsymbol{S}^1 = \boldsymbol{S}$, we repeat the subsequent steps for each $t = 1, 2, \dots, r$:

$$\boldsymbol{A}^t = \sigma(\boldsymbol{S}^t / \tau), \tag{6}$$

$$s_{ij}^{t+1} = s_{ij}^t + \log(1 - \sum_{j \in \mathbb{B}} a_{ij}^t), \tag{7}$$

where $\sigma$ denotes the global softmax function and $\tau > 0$ represents a temperature scale that regulates the relaxation. In each step $t$, this process computes the $\boldsymbol{A}^t \in [0, 1]^{|\mathbb{A}| \times |\mathbb{B}|}$ with $\sum_{i \in \mathbb{A}, j \in \mathbb{B}} a_{ij} = 1$, representing the soft-argmax. Subsequently, the similarity $\boldsymbol{S}^t$ is updated to suppress the entire outbounding edges from the softly selected nodes in $\mathbb{A}$ by Eq. 6. Afterward, the soft adjacency matrix $\tilde{\boldsymbol{E}}$ is defined as follows:

$$\tilde{e}_{ij} = \frac{a_{ij}^*}{\max(1, \text{sg}(\sum_{j \in \mathbb{B}} a_{ij}^*))}, \quad \text{where} \quad \boldsymbol{A}^* = \sum_{t=1}^{r} \boldsymbol{A}^t, \tag{8}$$

with $\sum_{ij} e_{ij} \leq r$, representing the total number of selected edges. The clipping function, composed of a max operator ($\max$) and stop-gradient ($\text{sg}$), is introduced to ensure that the resulting $\tilde{\boldsymbol{E}}$ is a valid continuous adjacency matrix.

Note that the soft grouping satisfies the aforementioned key conditions. As $\tau \to 0$, $\boldsymbol{A}^t$ converges to the one-hot matrix that indicates the nodes in $\mathbb{A}$ and $\mathbb{B}$ with maximum $s_{ij}^t$. This results in $\boldsymbol{A}^*$ representing $r$ most similar pair, thus satisfying the first condition. Meanwhile, edges associated with such nodes in $\mathbb{A}$ are excluded from future selection according to Eq. 7, since $\log(1 - \sum_{j \in \mathbb{B}} a_{ij}^t) \to -\infty$, ensuring at most one selection per node in $\mathbb{A}$. This holds true even in the non-asymptotic case, as the clipping function guarantees $\sum_{j \in \mathbb{B}} \tilde{e}_{ij} \leq 1$, thereby meeting the second condition.

### 3.3 Soft Merging

While the soft grouping and the resulting soft adjacency matrix effectively approximate the grouping process, it is crucial to design the merging operator to incorporate such soft decisions. In our approach, for the given soft adjacency matrix where $\tilde{e}_{ij}$ corresponds to tokens $i \in \mathbb{A}$ and $j \in \mathbb{B}$, our soft merging is designed such that the $i$th token merges into the $j$th token in proportion to the value of $\tilde{e}_{ij}$.

Our soft merging operator applies the asynchronous updates on tokens in two sets, $\mathbb{A}$ and $\mathbb{B}$. For each token $j \in \mathbb{B}$, the operator update their feature $\boldsymbol{x}_j$ and the effective size $m_j$ by aggregating tokens in $\mathbb{A}$ based on the soft adjacency matrix $\tilde{\boldsymbol{E}}$ from Section 3.2 by:

$$\hat{\boldsymbol{x}}_j \leftarrow \frac{m_j \boldsymbol{x}_j + \sum_{i \in \mathbb{A}} \tilde{e}_{ij} m_i \boldsymbol{x}_i}{m_j + \sum_{i \in \mathbb{A}} \tilde{e}_{ij} m_i}, \quad \hat{m}_j \leftarrow m_j + \sum_{i \in \mathbb{A}} \tilde{e}_{ij} m_i. \tag{9}$$

One the other hand, for each token $i \in \mathbb{A}$, the operator update only its effective size $m_i$ while maintaining the feature:

$$\hat{m}_i \leftarrow m_i(1 - \sum_{j \in \mathbb{B}} \tilde{e}_{ij}). \tag{10}$$

Note that with the binary adjacency matrix $\boldsymbol{E}'$, the effective size of the tokens in $\mathbb{A}$ reduces to zero by Eq. 10 if they have outbounding edges. Such tokens will be excluded from the subsequent merging process by Eq. 9. This process is simulated continuously during training with our soft adjacency matrix (*i.e.*, each token will be continuously absorbed into others), while it is used to actually reduce the tokens at inference using a discretized adjacency matrix.

Interestingly, we observe that the decoupled embedding, trained with soft merging at a high reduction rate $r$, generalizes well to lower rates $r' \leq r$. This is presumably because the decoupled embedding is learned to sort $r$ most similar token pairs by the relaxed top-k operator (Eqs. 6, 7), thereby including the sorting for smaller reduction rates $r' \leq r$.

### 3.4 Training and Inference

**Training**   Thanks to *decoupled* embedding modules, training can be conducted in two distinct ways: modular and end-to-end training. In modular training, we train only the embedding modules while keeping the ViT parameters frozen. This allows our method to fully leverage off-the-shelf models while effectively adapting only the merging policy to each task. In end-to-end training, we jointly train all ViT parameters along with our embedding modules. Since our continuous merging operators do not reduce the number of tokens during training, we alternate updates between the embedding layers and ViT parameters to save computation. Specifically, when updating the ViT parameters, we fix the embedding layers and use the discretized grouping and merging operators, which allows token reduction in the ViT forward pass, greatly enhancing the efficiency. Conversely, when updating the embedding modules, we apply the soft grouping and merging operators while fixing the ViT parameters. We alternate this procedure with much more frequency on ViT updates, since the embedding layers have considerably smaller parameters ($\approx$1%) and hence quickly converge. For both modular and end-to-end training, we simply train our method to minimize the task loss.

**Inference**   For inference, we discretize the continuous operators in the grouping and merging processes, and perform the hard token merging utilizing the learned decoupled embeddings. As explained in Sec. 3.2 and Sec. 3.3, our soft grouping and merging modules are asymptotically equivalent to BSM of ToMe. Consequently, we employ BSM to speed up the inference.

## 4 Experiments

We apply our method, DTEM, for token merging in image classification, captioning, and segmentation. In Sec. 4.1, we first evaluate our method in the ImageNet-1k [8] classification with two setups:

Table 1: **Classification results with off-the-shelf frozen pre-trained models**. Reduction roughly represents the decreases in FLOPs.

| Reduction | Method | Acc@1 | GFLOPs | im/s | Method | Acc@1 | GFLOPs | im/s |
|---|---|---|---|---|---|---|---|---|
| - | DeiT-S [30] | 79.83 | 4.64 | 1390 | DeiT-B [30] | 81.79 | 17.7 | 440 |
| 35% | EViT [18] | 78.50 | 3.03 | 2069 | EViT [18] | 80.45 | 11.6 | 658 |
|  | ToMe [2] | 79.12 | 3.02 | 1917 | ToMe [2] | 80.57 | 11.5 | 628 |
|  | **DTEM** | **79.44** | 2.91 | 1991 | **DTEM** | **81.01** | 11.1 | 653 |
| 50% | EViT [18] | 74.10 | 2.33 | 2672 | EViT [18] | 75.11 | 8.9 | 854 |
|  | ToMe [2] | 78.01 | 2.32 | 2457 | ToMe [2] | 77.92 | 8.82 | 823 |
|  | **DTEM** | **78.99** | 2.35 | 2430 | **DTEM** | **79.54** | 8.88 | 818 |
| - | MAE-B [11] | 83.72 | 17.7 | 438 | MAE-L [11] | 85.95 | 61.8 | 131 |
| 35% | EViT [18] | 82.11 | 11.7 | 658 | EViT [18] | 85.22 | 42.4 | 189 |
|  | ToMe [2] | 82.33 | 11.5 | 628 | ToMe [2] | 85.46 | 42.5 | 186 |
|  | **DTEM** | **82.80** | 11.6 | 653 | **DTEM** | **85.61** | 42.9 | 185 |
| 50% | EViT [18] | 75.95 | 8.9 | 854 | EViT [18] | 82.77 | 33 | 244 |
|  | ToMe [2] | 78.88 | 8.82 | 823 | ToMe [2] | 84.21 | 31.1 | 252 |
|  | **DTEM** | **80.37** | 8.88 | 818 | **DTEM** | **84.68** | 31.4 | 250 |

*modular* and *end-to-end* training. We further present our results on COCO [6] image captioning in Sec. 4.2 and ADE20K [41] semantic segmentation in Sec. 4.3 to demonstrate that our method can be applied to tasks requiring various levels of granularity in representation. We then provide a series of analyses on the importance of decoupled embedding, the design choices of embedding module, and data/training efficiency, complemented by visualizations, in Sec. 4.4.

## 4.1 Image Classification

**Setup** We conducted an image classification experiment on ImageNet-1k [8] dataset with 1.28M training and 50k validation images. We apply our method and baselines to various pre-trained ViT models, including DeiT-S/B [30], MAE-B/L [11], and LV-ViT-S [13]. The image resolution in training and testing is $224 \times 224$ unless otherwise stated. We also present the results for DeiT-T and AugReg [28] ViT-S (with a resolution of $384 \times 384$) in the supplementary material (A.1). We report top-1 accuracy (Acc@1) on the validation set, with floating-point operation (FLOPs) and throughput (images per second, im/s) to quantify the computation reduction. For the throughput, we measured on a single NVIDIA 3090 GPU with a batch size of 128 and fp32.

**Implementation detail** We mostly follow the fine-tuning setup from [26, 18], which is based on the training recipe of DeiT [30]. We initialize the ViTs with pre-trained weights and train for 30 epochs, as in most baselines [26, 18, 34, 10]. For token reduction, we employ a uniform reduction strategy, where the *reduction rate* $r$ represents the number of tokens removed in each transformer block. When training the embedding modules, we apply the reduction rate $r = 16$ to ViT-S/B models and $r = 8$ to the ViT-L model. As the embedding module, we use a linear layer with an output dimension of $64$ for ViT-S/B and $128$ for ViT-L. We use a temperature scale of $\tau = 0.1$ and also scale the similarity by $0.1$ prior to soft grouping. More details can be found in the supplementary material A.2.

**Modular training** Table 1 reports classification results when approximately 35% and 50% of FLOPs are reduced by applying token reduction methods to frozen pre-trained ViTs, with ViT parameters remaining unchanged. We compare DTEM with ToMe [2] and EViT [18] in this setting. The results demonstrate that DTEM consistently outperforms the baselines. Specifically, with a 35% reduction in FLOPs, our method improves performance by +0.15% to +0.47% compared to ToMe across all DeiT-S/B and MAE-B/L models. For a reduction of 50% in FLOPs, DTEM significantly improves performance by +0.47% to +1.64%, while adding less than 1% additional FLOPs.

In Table 2, we further applied our method to LV-ViT [13], a variant of standard ViT. LV-ViT employs an input embedding module consisting of convolution layers to better tokenize the input image. We apply token merging into the first 12 transformer blocks of LV-ViT-S. Consistent with previous results, DTEM achieves a +0.2% accuracy gain over ToMe, demonstrating its applicability to LV-ViT. Notably, despite optimizing only the added embedding module parameters, this performance is comparable to other state-of-the-art methods [18, 34] that fully fine-tune the ViT parameters.

Table 2: **Classification results with LV-ViT-S**. * indicates the results with off-the-shelf frozen pretrained model.

| Method | Acc@1 | GFLOPs | im/s |
|---|---|---|---|
| LV-VIT-S [13] | 83.3 | 6.6 | 879 |
| EViT [18] | **82.5** | 3.9 | - |
| eTPS [34] | **82.5** | 3.8 | - |
| ToMe* [2] | 82.3 | 3.69 | 1574 |
| **DTEM*** | **82.5** | 3.73 | 1571 |

Figure 2: **Classification results under different FLOPs and throughputs**. All methods are end-to-end trained.

Table 3: **Comparison of classification results with prior arts**. † denotes the baseline results implemented by ourselves.

| Method | Acc@1 | GFLOPs | im/s |
|---|---|---|---|
| *Comparison with DeiT-S* | | | |
| DyViT [26] | 79.3 | 2.9 | 2082 |
| Evo-ViT [38] | 79.4 | 3 | 2031 |
| EViT† [18] | 79.51 | 3 | 2069 |
| ToMe† [2] | 79.68 | 3 | 1917 |
| ATS [10] | 79.7 | 2.9 | - |
| BAT [20] | 79.6 | 3 | - |
| eTPS [34] | 79.7 | 3 | - |
| **DTEM** | **79.85** | 2.9 | 1991 |
| DyViT [26] | 78.5 | 2.5 | 2429 |
| EViT† [18] | 78.63 | 2.3 | 2672 |
| BAT [20] | 79.0 | 2.3 | - |
| eTPS [34] | 79.2 | 2.3 | - |
| ToMe† [2] | 79.25 | 2.3 | 2457 |
| **DTEM** | **79.38** | 2.3 | 2430 |
| *Comparison with DeiT-B* | | | |
| DyViT [26] | 81.3 | 11.6 | 657 |
| Evo-ViT [38] | 81.3 | 11.6 | - |
| EViT [18] | 81.3 | 11.6 | 658 |
| ToMe† [2] | 81.37 | 11.5 | 628 |
| **DTEM** | **81.60** | 11.6 | 624 |
| **DTEM** | **81.47** | 11 | 653 |
| EViT [18] | 80.0 | 8.9 | 854 |
| ToMe† [2] | 80.58 | 8.8 | 823 |
| **DTEM** | **80.74** | 8.9 | 818 |

**End-to-end training**  Figure 2 depicts the classification results under different FLOPs and throughputs when token reduction methods are applied through end-to-end training. We compared our method with a fine-tuned version of ToMe [2] and EViT [18]. For the baselines, we report accuracies by training each model on specific target computation demands under varying reduction rates, *e.g.*, $r = \{16, 13, 12, 11\}$ for ToMe. Conversely, for DTEM, we train a single model with a reduction rate of $r = 13$ for fine-tuning the ViT parameters, while maintaining $r = 16$ when training the embedding module.[1] We then adjust the reduction rate and report the corresponding accuracies during inference.

The results demonstrate that our method consistently outperforms the baselines across all levels of computational reduction. Specifically, our method surpasses the baseline accuracy by 0.12% to 0.2% in DeiT-S while adding a small amount of FLOPs and degradation in throughput. This leads to an improved trade-off between accuracy and computational resources, such as FLOPs and throughput. A notable aspect of DTEM is its ability to provide a single trained model that is generalized across various reduction rates. This can mitigate the training and storage costs associated with the multiple rounds of full fine-tuning often required to support different levels of computational reduction.

**Comparison to State-of-The-Art**  While the results in Figure 2 verify the effectiveness of our method in end-to-end training, we compare DTEM's performance with more token reduction methods in Table 3. We mainly considered the 30 epochs training results used in [26, 18, 34, 10] for a fair comparison.[2] The table shows that our method achieves superior accuracy compared to other prior arts when computational costs are equated.

### 4.2 Image Captioning

To demonstrate the broad applicability of our method, we apply DTEM to image captioning, a task extensively studied in the vision-language domain. Recent captioning models with ViTs typically utilize all output patch features to ensure the caption generation is grounded in richer and more

---

[1]As explained in Sec. 3.4, we alternate updates between the embedding modules and ViT parameters for end-to-end training.

[2]We further include comparison results from 100 epochs of training in the supplementary material A.1, demonstrating superior performance.

Table 4: **Image captioning evaluation results when token merging is applied**. We report with caption evaluation metrics: BLEU-4 (B@4), CIDEr (C), METEOR (M) and SPICE (S). Reduction represents the decreases in FLOPs within the ViT encoder, and # indicates the number of tokens passed to language decoder.

| | Reduction | B@4 | M | C | S | # | | Reduction | B@4 | M | C | S | # |
|---|---|---|---|---|---|---|---|---|---|---|---|---|---|
| GIT-B [33] | - | 38.8 | 30.1 | 127.6 | 23.6 | 197 | GIT-L [33] | - | 40.7 | 29.6 | 134 | 23.8 | 197 |
| ToMe [2] | 32% | 34.6 | 26.4 | 113.1 | 20.3 | 77 | ToMe [2] | 31% | 36.9 | 27.3 | 122.1 | 21.5 | 77 |
| | 35% | 33.5 | 25.8 | 109.3 | 19.8 | 65 | | 37% | 36.4 | 27.1 | 120.1 | 21.5 | 53 |
| | 38% | 33.3 | 25.5 | 107.9 | 19.5 | 53 | | 43% | 34.0 | 25.8 | 112.2 | 20.2 | 29 |
| | 41% | 31.9 | 24.8 | 104.3 | 19.0 | 41 | | 49% | 31.7 | 24.8 | 105.1 | 19.3 | 7 |
| DTEM | 31% | 36.2 | 27.1 | 118.1 | 20.8 | 77 | DTEM | 31% | 37.9 | 27.8 | 124.4 | 21.9 | 77 |
| | 34% | 34.5 | 26.5 | 114.2 | 20.5 | 65 | | 37% | 37.0 | 27.5 | 122.9 | 21.7 | 53 |
| | 37% | 34.3 | 26.2 | 112.9 | 20.1 | 53 | | 43% | 35.7 | 26.6 | 117.6 | 20.9 | 29 |
| | 41% | 33.3 | 25.7 | 110.4 | 19.9 | 41 | | 49% | 33.3 | 25.7 | 111.1 | 20.1 | 7 |

Table 5: **Results on semantic segmentation when token merging is applied**. The reduction ratio indicates the portion of merged tokens.

| Model | Method | | Baseline $(r=0)$ | Reduction ratio | | | |
|---|---|---|---|---|---|---|---|
| | | | | 0.2 | 0.3 | 0.4 | 0.5 |
| Seg-S-Mask/16 [29] | ToMe [2] | GFLOPs | 36.28 (100%) | 30.82 (85.0%) | 27.18 (74.9%) | 23.7 (65.3%) | 20.46 (56.4%) |
| | | mIoU | 45.28 | 44.88 | 43.98 | 41.70 | 36.62 |
| | Ours | GFLOPs | 36.28 (100%) | 29.39 (81%) | 25.75 (71%) | 22.27 (61.4%) | 19.03 (52.5%) |
| | | mIoU | 45.28 | **44.96** | **44.3** | 42.64 | **38.92** |

detailed information about the image, which is crucial for accurate captioning. However, using all patches may be inefficient due to redundancy in image tokens, motivating the use of token merging.

**Setup**  We experiment with the COCO [6] caption dataset using the train/val/test split from [14]. We use COCO fine-tuned GIT [33] models, each consisting of a ViT-B or L image encoder and a language decoder. The embedding modules are trained modularly with a language modeling loss and use the best model identified through cross-validation. The quality of captions is evaluated using metrics of BLEU-4 [24], METEOR [22], CIDEr [32], and SPICE [1], while the computational cost is reported in terms of the ViT encoder's floating-point operations (FLOPs) and the number of tokens (#) passed to the language decoder. More details are provided in the supplementary material A.2.

**Results**  Table 4 presents the image captioning results on the test split when DTEM or ToMe [2] are applied. DTEM outperforms ToMe, achieving a better trade-off between captioning quality and computation cost. Specifically, with the GIT-B model, DTEM enhances the CIDEr score by $+5.0$ to $+6.0$ across reductions in FLOPs of $31\%$ to $41\%$. Similarly, we observe an improvement ranging from $+2.3$ to $+6.0$ with the GIT-L model. These results confirm that DTEM provides a better set of patch representations by effectively summarizing the information in the image tokens.

### 4.3 Semantic Segmentation

To further demonstrate DTEM's applicability, we apply our method to semantic segmentation, a widely studied computer vision task with numerous applications.

**Setup**  We use a pre-trained Segmenter [29] model and evaluate the token merging on the ADE20K [41] dataset, which contains 25k training data across 150 fine-grained semantic categories. Unlike image classification or captioning tasks, segmentation models—including the Segmenter—require complete image patches (tokens) in the end to decode the segmentation mask. To address this, we follow the approach proposed in [3] that repeatedly merges tokens before each component (*e.g.*, self-attention and feed-forward network) and then un-merges them after processing the component. We modularly trained our decoupled embedding modules using the cross entropy loss. We report mean intersection-over-union (mIoU) and floating-point operations (FLOPs) for performance and computational cost, respectively. More implementation details are provided in the Appendix A.2.

Table 6: **Ablation study on the impact of decoupled embedding**. We successively add *soft token merging* and *decoupled embedding module* into ToMe. The number in parentheses indicates the reduction in FLOPs.

| Arch. | Method | Acc. (-35%) | Acc. (-50%) |
|---|---|---|---|
| | ToMe [2] | 79.68 | 79.25 |
| DeiT-S | + *soft token merging* | 79.57 | 79.15 |
| | + *decoupled embedding* | **79.85** | **79.38** |
| | ToMe [2] | 81.37 | 80.58 |
| DeiT-B | + *soft token merging* | 81.48 | 80.49 |
| | + *decoupled embedding* | **81.60** | **80.74** |

Table 7: **Kendall rank correlation coefficient changed through training**. We report changes in the Kendall rank correlation between token similarities derived from two different features: self-attention keys and decoupled embedding.

| | 1 to 4 blocks | 5 to 8 blocks | 9 to 12 blocks |
|---|---|---|---|
| Before training | 0.517 | 0.457 | 0.591 |
| After training | 0.401 | 0.402 | 0.519 |

Table 8: **Comparison with alternative design choices for soft grouping**. As an alternative, we experiment with applying the Gumbel Softmax (GS) to replace the top-1 operation from ToMe, enabling differentiation.

| Method | Acc. (-35%) | Acc. (-50%) |
|---|---|---|
| DynamicViT [26] (off-the-shelf) | 78.96 | 75.53 |
| ToMe [2] + GS + soft merging | 79.2 | 78.14 |
| DTEM | **79.44** | **78.99** |
| - without prop attn | 79.16 | 77.86 |

Figure 3: **Ablation study on decoupled embedding module design**: (a) decoupled embedding dimension and (b) number of hidden layers.

**Results**  Table 5 reports the semantic segmentation results when token merging methods, *i.e.*, ToMe [2] and DTEM, are applied to Segmenter with ViT-S. Our method consistently offers a better mIoU to GFLOPs trade-off compared to ToMe. Specifically, DTEM achieves a +0.32 to +1.3 improvement in mIoU over ToMe when 25% to 50% FLOPs are reduced. These results demonstrates the applicability of DTEM in semantic segmentation for enhancing token merging.

## 4.4   Analysis

We conduct an analysis of DTEM in ImageNet-1k classification, specifically within a modular training setting using the DeiT-S model, unless if otherwise stated.

**Importance of decoupled embedding**  In Table 6, we ablate the impact of decoupled embedding in end-to-end training. We observe that naive integration of soft grouping and merging applied to the keys of self-attention, as in ToMe [2], degrades the performance. This confirms that decoupled embedding is crucial in DTEM to provide more compelling features for token merging. In Table 7, we further investigate whether the decoupled embedding used for merging diverges from the intermediate features after training. We report the Kendall rank correlation between token similarities derived from two different features—self-attention keys and decoupled embedding—before and after training. The results show a decreased correlation after training, indicating that the decoupled embedding learns a different feature for token merging, distinct from the intermediate features.

**Design choices for soft grouping**  In Table 8, for the analysis of soft grouping, we compared several approaches: (1) as the pruning baseline, we tested DynamicViT [26] applied to a frozen ViT, (2) integrating Gumbel Softmax (GS) to replace the top-1 operation from ToMe to enable differentiation, (3) our method, and (4) our method without proportional attention. The results indicate that our proposed design for soft grouping performs the best, with proportional attention proving to be crucial.

**Embedding module design**  Figure 3 (a) and (b) show the impact of different embedding dimensions and the number of hidden layers in the embedding module, respectively. In the main results (Sec. 4.1), we use the embedding dimension of 64 and the module without hidden layers. We observe that a simple affine transformation offers sufficient gain while keeping computational costs low.

**Effect of reduction rate on training**  In Table 9, we analyze the effect of the reduction rate used during training. We observe that the decoupled embedding module, when trained with a high reduction rate $r$, generalizes well to lower rates during inference. Therefore, it is generally sufficient to set the reduction rate $r$ during training to the maximum number of tokens we wish to reduce during inference.

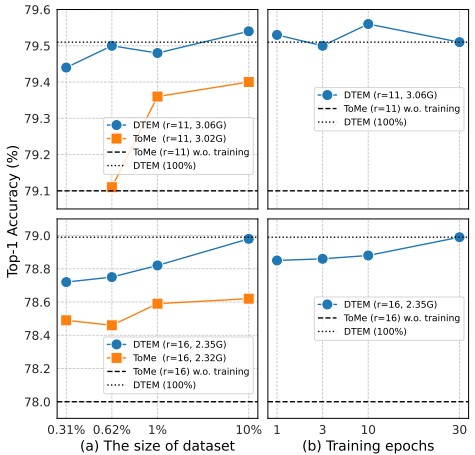

(a) The size of dataset     (b) Training epochs

Figure 4: **Image classification results on data and train efficiency**: (a) dataset size and (b) training epochs. In the experiments, DTEM is modularly trained on DeiT-S model, while ToMe undergoes end-to-end training.

Table 9: **Effect of the reduction rate $r$ in soft grouping**. Results style: **best**, second best.

| Reduction rate $r$ at training | Reduction rate $r$ at inference | | | |
|---|---|---|---|---|
| | 16 | 14 | 12 | 10 |
| 16 | **78.92** | **79.33** | 79.42 | 79.60 |
| 14 | 78.80 | 79.23 | **79.43** | 79.60 |
| 12 | 78.77 | 79.22 | 79.41 | **79.61** |
| 10 | 78.67 | 79.19 | 79.38 | 79.60 |

Figure 5: **Visualization of merged tokens**. We apply a reduction rate $r = 16$, leading to 11 merged tokens in the final output.

**Data/Train efficiency** In Figure 4 (a) and (b), we examine the data and training efficiency of DTEM with modular training, respectively. Owing to the parameter-efficient modular approach, DTEM improves the performance of merging even with a limited amount of training data and epochs. Specifically, in Figure 4 (a), our method achieves a gain of +0.44% accuracy even when trained with 4000 images, equaling 0.31% of the total dataset. DTEM outperforms the end-to-end training approach with ToMe under 10% of the total dataset for both 35% and 50% reduction of FLOPs, highlighting the potential benefit of our modular training when the entire dataset is unavailable. Moreover, in Figure 4 (b), the results on the effect of varying training epochs show that DTEM quickly converges even at the first epoch. Since DTEM employs modular training even in end-to-end learning by the alternative optimizations (Sec. 3.4), such rapid convergence is also useful in reducing the cost of end-to-end training.

**Visualization** In Figure 5, we visualize the token merging to compare ToMe [2] and our modularly trained DTEM. Tokens belonging to the same group are color-coded identically, highlighting the grouping changes induced by the decoupled embedding. DTEM prioritizes merging background tokens, allocating more tokens to foreground objects. In contrast, ToMe allocates more tokens to the background, indicating a less focused approach to foreground objects.

## 5 Conclusion

We propose Decoupled Token Embedding for Merging (DTEM) that improves token merging via decoupled token embedding derived directly from the token merging process. Our method introduces the decoupled embedding, learned through our continuously relaxed token merging, to exploit dedicated features for token merging. The decoupled embedding enhances token merging by resolving the dependency of token merging on intermediate features and enables modular training, effectively utilizing the frozen pre-trained models. We experiment with DTEM on classification, captioning, and segmentation using various pre-trained ViT models. The experimental results demonstrate that our method consistently improves token merging, highlighting the importance of features tailored specifically for token merging.

## Acknowledgments and Disclosure of Funding

This work was in part supported by the National Research Foundation of Korea (RS-2024-00351212 and RS-2024-00436165), and Institute of Information & communications Technology Planning & Evaluation (IITP) grant (RS-2022-II220926, RS-2024-00509279, RS-2021-II212068, and RS-2019-II190075) funded by the Korean government (MSIT).

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

# A  Appendix

## A.1  More Results

**Full Classification Results**   Table 10 and Table 11 report the full image classification results of our method with DeiT-S/B [30] and MAE-B/L [11], respectively. For DeiT, we report the full results of both modular and end-to-end training settings. For MAE, we report full results in the modular training setting.

Table 10: Full image classification results of our method with DeiT-S/B models at different reduction rates $r$.

| Model | GFLOPs | $r$ | Top-1 Accuracy Modular | Top-1 Accuracy End-to-end |
|---|---|---|---|---|
| DeiT-S | 2.35 | 16 | 78.99 | 79.38 |
|  | 2.48 | 15 | 79.17 | 79.61 |
|  | 2.63 | 14 | 79.20 | 79.60 |
|  | 2.77 | 13 | 79.49 | 79.74 |
|  | 2.91 | 12 | 79.44 | 79.85 |
|  | 3.06 | 11 | 79.51 | 79.82 |
| DeiT-B | 8.88 | 16 | 79.54 | 80.74 |
|  | 9.40 | 15 | 80.03 | 81.03 |
|  | 9.95 | 14 | 80.35 | 81.17 |
|  | 10.50 | 13 | 80.68 | 81.37 |
|  | 11.05 | 12 | 81.01 | 81.47 |
|  | 11.60 | 11 | 81.01 | 81.60 |

Table 11: Full image classification results of our method with MAE-B/L models at different reduction rates $r$. Only the embedding modules are modularly trained.

| Model | GFLOPs | $r$ | Top-1 Accuracy |
|---|---|---|---|
| MAE-B | 8.88 | 16 | 80.37 |
|  | 9.40 | 15 | 81.29 |
|  | 9.95 | 14 | 81.82 |
|  | 10.50 | 13 | 82.24 |
|  | 11.05 | 12 | 82.56 |
|  | 11.60 | 11 | 82.80 |
| MAE-L | 31.42 | 8 | 84.68 |
|  | 35.21 | 7 | 84.92 |
|  | 39.05 | 6 | 82.58 |
|  | 42.90 | 5 | 85.61 |

**Extended image captioning evaluation results**   In Table 12, we report Figure 4 results across a broader reduction range. The result shows that our method is particularly effective in challenging, more resource-constrained settings with higher reduction rates. caption. We note that for reduction rates over 41% and 49% for GIT-B and GIT-L respectively, there was a significant decrease in captioning quality.

Table 12: Full image captioning evaluation results when token merging is applied.

|  | Reduction | B@4 | M | C | S | # |  | Reduction | B@4 | M | C | S | # |
|---|---|---|---|---|---|---|---|---|---|---|---|---|---|
| GIT-B | - | 38.8 | 30.1 | 127.6 | 23.6 | 197 | GIT-L | - | 40.7 | 29.6 | 134 | 23.8 | 197 |
| ToMe | 12% | 37.9 | 28.6 | 123.7 | 22.4 | 149 |  | - | - | - | - | - | - |
|  | 25% | 35.4 | 27.1 | 115.9 | 21 | 101 |  | 18% | 40.1 | 28.9 | 131.1 | 23 | 125 |
|  | 27% | 35.7 | 26.9 | 115.3 | 20.9 | 89 |  | 24% | 39.4 | 28.8 | 128.7 | 22.7 | 101 |
|  | 32% | 34.6 | 26.4 | 113.1 | 20.3 | 77 | ToMe | 31% | 36.9 | 27.3 | 122.1 | 21.5 | 77 |
|  | 35% | 33.5 | 25.8 | 109.3 | 19.8 | 65 |  | 37% | 36.4 | 27.1 | 120.1 | 21.5 | 53 |
|  | 38% | 33.3 | 25.5 | 107.9 | 19.5 | 53 |  | 43% | 34.0 | 25.8 | 112.2 | 20.2 | 29 |
|  | 41% | 31.9 | 24.8 | 104.3 | 19.0 | 41 |  | 49% | 31.7 | 24.8 | 105.1 | 19.3 | 7 |
| DTEM | 12% | 38 | 28.6 | 124.2 | 22.3 | 149 |  | - | - | - | - | - | - |
|  | 25% | 36 | 27.3 | 118.9 | 21.4 | 101 |  | 18% | 40.1 | 29.1 | 131.5 | 23.2 | 125 |
|  | 27% | 36.4 | 27.4 | 119.3 | 21.4 | 89 |  | 24% | 39.4 | 28.9 | 129.5 | 23 | 101 |
|  | 31% | 36.2 | 27.1 | 118.1 | 20.8 | 77 | DTEM | 31% | 37.9 | 27.8 | 124.4 | 21.9 | 77 |
|  | 34% | 34.5 | 26.5 | 114.2 | 20.5 | 65 |  | 37% | 37.0 | 27.5 | 122.9 | 21.7 | 53 |
|  | 37% | 34.3 | 26.2 | 112.9 | 20.1 | 53 |  | 43% | 35.7 | 26.6 | 117.6 | 20.9 | 29 |
|  | 41% | 33.3 | 25.7 | 110.4 | 19.9 | 41 |  | 49% | 33.3 | 25.7 | 111.1 | 20.1 | 7 |

**Classification Results with DeiT-T**   In Table 13, we further report the classification results of our method with DeiT-T, demonstrating that our method consistently improves token merging in modular and end-to-end training settings. We follow the 30 epoch training settings of DeiT-S/B, except that we remove the MixUp and CutMix augmentations, as the performance improves without them.

**Training for Longer Epochs**   In Table 14, we further report comparison results from longer training epochs. We note that some methods [18, 34] can improve their performance by training for significantly more epochs, *e.g.*, 100 epochs. To make a fair comparison, we also trained DTEM for 100 epochs, as with the baseline, *i.e.*, dTPS [34]. The results show that DTEM outperforms the baselines in DeiT-T and is comparable to dTPS in DeiT-S, while incurring much lower training costs. This efficiency is credited to DTEM's ability to effectively generalize across various reduction

Table 13: Full image classification results of our method with DeiT-T at different reduction rates $r$. () indicates the improvement in accuracy over ToMe [2].

| GFLOPs | $r$ | Top-1 Accuracy | |
|---|---|---|---|
| | | Modular | End-to-end |
| 0.65 | 16 | 70.20 (+1.52) | 72.16 (+0.26) |
| 0.69 | 15 | 70.73 (+1.42) | 72.51 |
| 0.73 | 14 | 71.04 (+1.08) | 72.57 |
| 0.77 | 13 | 71.27 (+0.74) | 72.77 |
| 0.81 | 12 | 71.36 (+0.50) | 72.88 |
| 0.85 | 11 | 71.51 (+0.47) | 72.94 (+0.18) |

Table 14: Image classification results with 100 epochs of end-to-end training.

| Model | Reduction | Method | Acc@1 |
|---|---|---|---|
| DeiT-T | 35% | dTPS (100ep) | 72.9 |
| | | DTEM (100ep) | **73.2** |
| | 50% | dTPS (100ep) | 72.1 |
| | | DTEM (100ep) | **72.4** |
| DeiT-S | 35% | dTPS (100ep) | **80.1** |
| | | DTEM (100ep) | 80.0 |
| | 50% | dTPS (100ep) | 79.7 |
| | | DTEM (100ep) | **79.7** |

rates with a single trained model, as detailed in Section 4.1. Note that dTPS requires multiple trained models to support different reductions in computation. Moreover, dTPS requires multiple training losses to achieve such performance, including the self-distillation loss, which brings the remaining tokens closer to those of the teacher model, and the KL divergence loss, which minimizes the difference in final predictions between the model and its teacher. In contrast, our method can achieve comparable or even better performance using only the task loss.

**Classification Results with AugReg ViT-S** In Table 15, we report experimental results with a larger number of tokens on image classification (AugReg [28] ViT-S with a resolution of $384 \times 384$), corresponding to smaller patches or higher input resolutions. The decoupled embedding module is trained modularly with a reduction rate of $r = 48$. The results show that our method can adapt to settings with an increased number of tokens, achieving performance gains.

Table 15: Image classification results with AugReg ViT-S pretrained on $384 \times 384$ resolution.

| Method | Reduction | Acc@1 | GFLOPs | im/s |
|---|---|---|---|---|
| ViT-S (384) | - | 83.8 | 15.7 | 394 |
| ToMe | 51.5% ($r = 47$) | 82.1 | 7.60 | 728 |
| DTEM | 52.0% ($r = 48$) | 82.3 | 7.54 | 733 |

**Effect of temperature scaling** In Table 16, we report classification results analyzing the effect of the temperature scaling. We experimented with a modular training using the DeiT-S model. We trained the decoupled embedding module for 10 epochs. We observe that values within the range of 0.1 to 0.3 consistently provide gains with an accuracy difference of 0.1%.

Table 16: Ablation study on the effect of temperature scaling.

| Temperature scale | Acc@1 (-50%) | Acc@1 (-35%) |
|---|---|---|
| 0.05 | 78.41 | 79.19 |
| 0.1 | 78.87 | 79.51 |
| 0.2 | 78.91 | 79.50 |
| 0.3 | 78.92 | 79.57 |
| 0.5 | 78.83 | 79.54 |
| 1 | 78.59 | 79.34 |

**End-to-End Training for Captioning** We further present the results of end-to-end training for token merging method applied to image captioning, using the GIT base model [33] on the COCO caption dataset. A separate learning rate of $0.0001$ is applied for updates to the embedding module, and $0.000002$ for the ViT and text decoder parameters. Our method is compared with a end-to-end trained version of ToMe [2] applied to the GIT base model. Both our method and ToMe are trained with a reduction rate of $r = 13$. The results are shown in Tab. 17. Although end-to-end training benefits both methods, our method offers a better trade-off between performance and computational efficiency. To be more specific, our method achieves an improvement of $+1.0$ to $+2.0$ in the CIDEr score.

Table 17: Image captioning evaluation results when token merging is applied with end-to-end training. We report with caption evaluation metrics: BLEU-4 (B@4), CIDEr (C), METEOR (M) and SPICE (S). # indicates the number of remaining tokens in the ViT output.

| Method | Reduction | B@4 | M | C | S | # tokens |
|---|---|---|---|---|---|---|
| ToMe | -32% | 37.8 | 28.0 | 122.0 | 21.6 | 77 |
| | -35% | 36.7 | 27.8 | 120.2 | 21.4 | 65 |
| | -38% | 36.7 | 27.3 | 118.8 | 21.0 | 53 |
| | -41% | 36.2 | 27.2 | 117.4 | 20.8 | 41 |
| DTEM | -31% | 37.9 | 28.1 | 123.3 | 21.9 | 77 |
| | -34% | 37.2 | 28.0 | 122.2 | 21.7 | 65 |
| | -37% | 36.8 | 27.6 | 120.2 | 21.2 | 53 |
| | -41% | 36.2 | 27.2 | 118.4 | 21.1 | 41 |

## A.2 Implementation Details

In this section, we provide a more detailed explanation of the implementation.

**Common details** During training, our method retains $N$ tokens, even if its effective size $m$ becomes zero. To ensure that these tokens do not participate in the relaxed merging process, we successively exclude the $r$ tokens with the minimum effective size from the relaxed merging at each transformer block. Consequently, after applying this exclusion process $l$ times, relaxed merging operates with $N - rl$ tokens at the $l$-th ViT block. We observe that blocking the gradient flow from the relaxed merging operators to the intermediate features of ViTs enhances task performance. Therefore, we use detached ViT intermediate features as input for the embedding module, allowing only the module to learn from the relaxed merging process. In end-to-end training, we alternate updates between the embedding module and the ViT parameters, as explained in (Sec. 3.4). We set for every 10 update cycles, the embedding layer is updated once, while the ViT parameters are updated nine times.

**Pre-trained models** For DeiT [30], we use TIMM [35] pre-trained models, which are available at `github.com/huggingface/pytorch-image-models`. For MAE [11], we use the ImageNet-1k fine-tuned checkpoints of the official implementation, at `github.com/facebookresearch/mae`. For LV-ViT-S [13], we use the official pre-trained model weights, available at `github.com/zihangJiang/TokenLabeling`. For image captioning using GIT [33], we use the COCO fine-tuned weights on HuggingFace [36], available at `huggingface.co/microsoft/git-base-coco`. For semantic segmentation using Segmenter [29], we use the Ade20K pre-trained checkpoint of the official implementation, at `github.com/rstrudel/segmenter`.

**Image Classification** Our method and baselines are trained for 30 epochs, each with a batch size of 1024. We use the AdamW optimizer with a cosine learning rate schedule and a weight decay of 0.0001. For image augmentation, we follow the DeiT training setting, using RandAug, MixUp, and CutMix. However, we omit MixUp and CutMix in DeiT-T and MAE training, as it performs better without them. For the learning rate of our method and the baselines, we conduct a hyperparameter search within $\{0.000001, 0.000005, 0.00001, 0.0001, 0.001\}$, selecting the learning rate that achieves the best performance. As a result, we use a learning rate of 0.0001 for modular training and 0.000005 for end-to-end updates of the ViT parameters, with a minimum learning rate set at 0.000001. Additionally, we implement a drop path rate of 0.1 only during end-to-end training when updating the ViT parameters. Regardless of whether the training is modular or end-to-end, we apply a reduction rate of $r = 16$ to train the embedding module with ViT-tiny/small/base models and $r = 8$ for ViT-Large models. For LV-ViT-S, we apply reduction to the first 12 transformer blocks. Meanwhile, to update the ViT parameters in end-to-end training, we use a different reduction rate of $r = 13$. We employ a temperature scale of $\tau = 0.1$ across different ViTs and also scale the similarity divided by 0.1 prior to the soft grouping operation.

**Image Captioning** In image captioning, we train and evaluate the models using LAVIS [16]. We train the GIT models over $20,000$ iterations, using a batch size of 256. We employ the AdamW optimizer along with a cosine learning rate schedule. The learning rate is set to 0.0001, gradually decreasing to 0.000001, with a weight decay of 0.0001. During training, the only form of image augmentation that we employ is resizing the images to $224 \times 224$; no other image augmentation techniques are used. For evaluation, beam search is applied with a beam size of 5, incorporating both repetition and length penalties set at 1. A reduction rate of $r = 13$ is used for the GIT base model,

and $r = 8$ for the GIT large model. We use the model with the best performance on the validation split and report its performance on the test split.

**Semantic Segmentation**    To apply ToMe and DTEM to the semantic segmentation task, we use the approach proposed in [3] that involves repeatedly merging tokens before each component, such as attention and the feed-forward network, and then un-merging them afterward. To be more specific, given a component of the ViT block, tokens are merged following the conventional token merging approach. For ToMe, we define the similarity between tokens using the input ViT features to the component. For DTEM, we define the similarity using the decoupled embeddings, which are produced by the decoupled embedding module that takes the same input as the component. After processing by the component, tokens are unmerged by duplicating the jointly processed tokens, as in [3]. Subsequently, the processed tokens are added with the residual connection.

We train and evaluate the model using MMSegmentation [7]. DTEM is trained over $40,000$ iterations with a batch size of $8$, following a 40k iteration schedule. During training, we use an image size of $512 \times 512$, and apply the default image augmentations, which include resize, random crop, random flip, and Photometric Distortion, as outlined in [29]. We remove 40% of the tokens at each component of the transformer block to train DTEM with Segmenter-ViT-S. We apply relaxed token merging when selecting the last $64$ tokens for merging, while the remaining tokens to be reduced are first selected through conventional discrete merging. We observe that this approach resolves unnecessary leakage from the clipping function and stabilizes training.

## A.3   Related Work

**Pruning tokens**    Early token reduction methods prune tokens based on their importance for task performance, keeping tokens in order of importance. The importance is often measured by a lightweight prediction/decision module [26, 39, 23] trained from task loss or using attention scores [18] to keep attentive tokens. The most widely used heuristic approach uses the class token attention scores as a metric of importance, as in [18, 20, 34, 19]. However, as mentioned in the Introduction, removing tokens results in information loss and degrades performance, especially when many tokens are reduced.

**Combining tokens**    A line of works attempts to combine tokens rather than removing them. The basic idea is to combine similar tokens, focusing on removing repetitive information to minimize the loss of information. However, early methods, which involved merging or grouping all tokens at certain stages, achieved limited success. PatchMerger [27] introduced a soft clustering approach using an attention-like module, while TokenPooling [21] used K-medoids clustering for cluster assignment to tokens. Recently, ToMe [2] has shown that token merging can be implemented efficiently by gradually merging a number of tokens at each transformer block with matching, resulting in promising performance even without training.[3] In addition, numerous studies focus on designing methods that consider the importance of individual tokens while addressing information loss by merging similar tokens. EViT [18] and Evo-ViT [38] propose to combine pruned tokens into a single token to address information loss, while BAT [20] and TPS [34] use the importance metric in conjunction with matching and clustering.

Despite their effectiveness, a common aspect among these methods is their reliance on intermediate features for merging tokens. We focus on improving usch embedding for merging with decoupled embedding learned directly through a continuously relaxed merging. Although not directly related, GroupViT [37] learns features through a grouping block, aiming to learn an implicit segmentation model without direct supervision. In contrast, our method aims to reduce the computation cost in the inference.

Additionally, DiffRate [5] proposes to learn the reduction strategy (or profile), which determines the number of tokens reduced at each transformer block, to enhance token reduction. We note that this approach is orthogonal to ours. While our method focuses on tailored features for merging and reduces a fixed number of tokens across blocks, DiffRate focuses on determining the number of tokens to be reduced in each block, relying on ViT's intermediate features for merging. We believe that jointly optimizing both components—the number of tokens reduced at each block and the features used for merging—could be a promising future direction.

---

[3]We note that our method can inherit the same advantage by employing identity mapping in decoupled layers.

### A.4 Limitations

Although our method conceptually appears to be applicable across a range of domains, including different modalities like text encoding, its effectiveness has so far been validated only in the computer vision tasks with ViTs. Another limitation is that, while our method enhances the computation time during inference, it does not reduce the computational cost during the training process. Devising a method that can sparsify the training process, thereby improving the overall computational efficiency in both the training and inference phases, is our future direction.

### A.5 Computing Resource

We do all our experiments on a GPU Server consists of Intel Xeon Gold 6230 CPU, 256GB RAM, and 8 NVIDIA RTX 3090 GPUs (with 24GB VRAM).

### A.6 More Visualization Results

In Fig. 6, we provide visualization results of token merging, i.e., ToMe and DTEM. we visualize the token merging by color coding the tokens belonging to the same group with identical colors, comparing ToMe [2] and modularly trained DTEM. The result highlights the difference in grouping induced by the decoupled embedding. DTEM prioritizes merging background patches and allocates more patches to foreground objects. Conversely, ToMe allocates more tokens to the background, indicating a less focused approach to foreground objects.

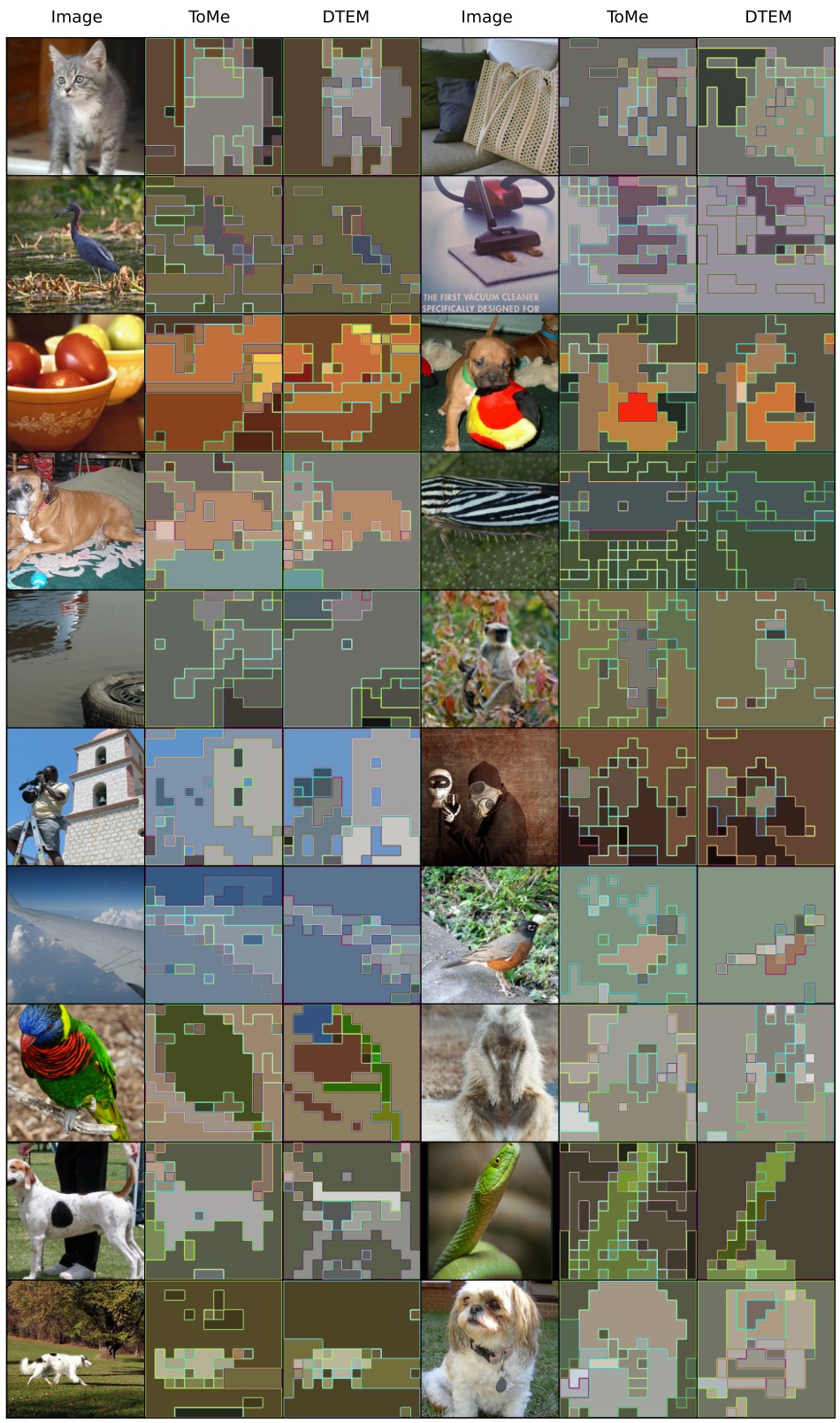

Figure 6: More visualization of merged tokens. We apply a reduction profile with $r = 16$, leading to 11 tokens remaining in the final output.

