# OpenReview forum: "Learning to Merge Tokens via Decoupled Embedding for Efficient Vision Transformers"
_NeurIPS.cc/2024/Conference — NeurIPS 2024 poster_

### Official Review · Reviewer_oGzP · 2024-07-11

**Soundness:** 3
**Presentation:** 3
**Contribution:** 2
**Rating:** 6
**Confidence:** 5

**Summary:**

This article provides a novel way to enhance the efficiency of token merging within ViTs. DTEM, the proposed method, introduces a lightweight embedding module that operates independently from the ViT's forward pass, overcoming the constraints imposed by utilizing intermediate features. DTEM can be integrate with existing ViT backbones and be trained either modularity by focusing solely on the decoupled embeddings or end-to-end by fine-tuning the entire network. The author shows that DTEM performs well in classification, captioning, and segmentation, demonstrating consistent improvements in token merging efficiency.

**Strengths:**

1.	The idea of decoupling seems novel for enabling the continuous relaxation of the token merging process, facilitating differential learning of the decoupled embeddings.
2.	The paper is clearly presented and easy to follow.
3.	Experimental results are promising. The method's applicability is demonstrated across multiple domains, including classification, captioning, and segmentation, illustrating its robustness and versatility.

**Weaknesses:**

1.	The model training process involves many hyperparameters, such as step in soft grouping, m in soft merging. Will the determination of these hyperparameters bring complexity to the model implementation? There is no ablation study on parameter m in soft merging.
2.	There is no ablation experiment in segmentation and caption to prove the effectiveness of the module on this task.

**Questions:**

1.	Though claimed decoupling, the overall process is similar to self-attention process especially in end-to-end training scenario. How to prove the decoupling effect except from the classification results?
2.	What are the main improvements of this method compared with ToMe? Does ToMe use the same soft grouping method? If not, it is not reflected in the ablation experiment.
3.	How to ensure a specific reduction rate?
4.	The comparison of soft grouping with other method such as Gumbel-Softmax in dynamic ViT[1].
[1 ] Rao Y, Zhao W, Liu B, et al. Dynamicvit: Efficient vision transformers with dynamic token sparsification[J]. Advances in neural information processing systems, 2021, 34: 13937-13949.

**Limitations:**

The authors have claimed its limitation for:
1.	Only in computer vision task
2.	Does not reduce the computational cost in the training process

---

> ### Author Rebuttal · Authors · 2024-08-07
>
> > W1. Will the determination of these hyperparameters bring complexity to the model implementation? There is no ablation study on parameter m in soft merging.
>
> &rarr; We report further analysis related to hyper parameters, regarding (1) the number of steps in soft grouping and (2) temperature scaling. We list the results in (Table S.1) and (Table S.2) of rebuttal pdf, respectively.
>
> For the number of steps (equals to the reduction rate during training) r in soft-grouping, we observe that the decoupled embedding module, when trained with a high reduction rate r, generalizes well to lower rates. Therefore, it is generally sufficient to set the number of steps to the maximum number of tokens we want to reduce during inference.
>
> For temperature scaling, we observed that values within the range of 0.1 to 0.3, tested with increments (0.05, 0.1, 0.2, 0.3, 0.5, 1.0), consistently provide gains with an accuracy difference of 0.1%.
>
> We clarify that the effective size m in soft merging is not a hyperparameter but a vector representing the sizes of combined tokens following the Equation 9. We also conducted experiments to determine if this effective size m and proportional attention (Equation 3) are necessary in our implementation by removing them. The result in (Table S.5. w.o. prop-attn) shows that both the effective size and the proportional attention based on it are crucial in our method.
>
> As a result, we believe that the hyperparameters in our method do not introduce significant complexity to our model implementation.
>
>
> > W2. There is no ablation experiment in segmentation and caption to prove the effectiveness of the module on this task.
>
> &rarr; Following the reviewer's comment, we further report ablation experiment results in captioning and segmentation to demonstrate the importance of the decoupled embedding module by varying the decoupled embedding dimension. The results (Table S.6) show that the decoupled embedding module indeed directly affects the quality of token merging. We also note that modular training is infeasible without the decoupled embedding module, which has external parameters independent from the ViT parameters.
>
> > Q1. How to prove the decoupling effect except from the classification results?
>
> &rarr; To this end, we further investigate whether the decoupled embedding used for merging diverges from the intermediate features as learning progresses. We monitor changes in the Kendall rank correlation between token similarities derived from two different features: self-attention keys (as in ToMe) and decoupled embeddings. The results in (Table S.4) show a decreasing correlation as learning progresses, indicating that the decoupled embeddings seek a different measure of similarity for merging, thereby verifying the benefits of decoupling.
>
> > Q2. What are the main improvements of this method compared with ToMe?
>
> &rarr; The main improvements of our method over ToMe can be summarized as follows:
>
> 1. Decoupled embedding for improved trade-off: Unlike ToMe, which uses intermediate ViT features for both encoding and token merging, our method introduces a decoupled embedding that distinctly separates the features used for token merging from those used for encoding. Our experimental results, detailed in (Table 3,4), demonstrate that this separation significantly enhances the performance and computational cost trade-offs by optimizing the merging policy independently.
> 2. Modular training on frozen ViTs: The decoupled embedding modules exist independently of the ViTs. This enables the improvement of token merging on top of frozen ViTs without altering the original ViT parameters. This modular approach is infeasible with ToMe, as it relies on ViT's intermediate features for merging, limiting its ability to enhance merging only through end-to-end training.
>
> Our ablation studies, shown in Table 4, explore the impacts of incorporating (1) soft grouping and merging operations, and (2) the decoupled embedding module into the ToMe. We also note that combining both components converges to our method. The results show that our approach, which integrates both components, achieves the best trade-off between performance and computational efficiency.
>
>
> > Q3. How to ensure a specific reduction rate?
>
> &rarr; As explained in W1, we observed that the decoupled embedding module, when trained with a high reduction rate r, generalizes effectively to lower rates. Therefore, during training, we set the number of steps (equivalent to the reduction rate) to the maximum number of tokens we aim to reduce during inference.
>
> During inference, we adjust the reduction rate as needed based on this generalization capability. Meanwhile, if there are specific targets for GFLOPs or throughput that need to be met, these can be achieved by iteratively adjusting the number of tokens merged at each ViT block, similar to the approach used in ToMe.
>
>
> > Q4. The comparison of soft grouping with other methods such as Gumbel-Softmax in [DynamicViT].
>
> &rarr; Some previous work in token pruning, such as DynamicViT, optimizes the selection of tokens to discard by employing a differentiable selection operator, such as the Gumbel-softmax. However, these approaches primarily focus on selecting individual tokens and thus are not directly applicable to token merging, which requires selecting pairs of tokens. A key contribution of our method is that we enable learning through token merging via our soft-grouping and merging operators.
>
> For the analysis on soft grouping, we compared several approaches: (1) integrating a Gumbel softmax with the top-1 operation from ToMe to enable differentiation, (2) applying DynamicViT to a frozen ViT setting (pruning), and (3) our method, which uses a modified relaxed top-k operation, in (Table S.5). The results indicate that our proposed method for soft grouping performs the best.

---

### Official Review · Reviewer_wYAw · 2024-07-11

**Soundness:** 3
**Presentation:** 3
**Contribution:** 3
**Rating:** 5
**Confidence:** 5

**Summary:**

This paper proposes DETM, which calculates similarity through additional embeddings instead of the original intermediate features. Additionally, it further introduces soft grouping and soft merging to make the merging process differentiable.

**Strengths:**

1. The paper is well-written and easy to follow.
2. The method is straightforward and consistently improves performance.
3. Comprehensive experiments demonstrate the effectiveness of the proposed method across different tasks, including image classification, image segmentation, and image captioning.

**Weaknesses:**

1. There is a lack of discussion and comparisons with recent related works, such as [1] and [2].
2. While the experiments show improved performance, I have concerns about the mechanism of introducing additional embeddings. The paper argues that original token merging is less effective because intermediate features are used for both encoding and merging. However, I do not believe using intermediate features directly is a drawback. The primary motivation of ToMe is to merge tokens with similar intermediate features. For example, if the similarity of two intermediate tokens is 1, merging these tokens is lossless, and additional embeddings are unnecessary in such scenarios. Therefore, the authors should provide more intuitive examples or an in-depth analysis beyond experimental comparisons to demonstrate the necessity of introducing additional embeddings.



[1] Diffrate: Differentiable compression rate for efficient vision transformers, ICCV 2023

[2] A Simple Romance Between Multi-Exit Vision Transformer and Token Reduction, ICLR 2024

**Questions:**

Please refer to the weaknesses section for detailed concerns.

**Limitations:**

The authors have discussed potential risks in the limitations section. Additionally, similar to other token reduction works, this study only conducts experiments on plain ViT architectures.

---

> ### Author Rebuttal · Authors · 2024-08-07
>
> > W1. There is a lack of discussion and comparisons with recent related works, such as [DiffRate] and [METR].
>
> &rarr; We appreciate your feedback regarding the need for discussions and comparisons with recent related works, specifically [DiffRate] and [METR].
>
> DiffRate focuses on determining the number of tokens to be reduced in each block, a concept that is orthogonal to our approach. Our method consistently reduces a fixed number of tokens across blocks, while focusing on tailored features for merging. In contrast, DiffRate relies on ViT's intermediate features for merging. Jointly optimizing both components—the number of tokens reduced at each block and features for merging—could be a promising future direction to enhance the efficiency and effectiveness of token reduction.
>
> On the other hand, METR addresses the inconsistency between CLS token attention and the importance of tokens in the early blocks of ViTs, particularly for CLS token-based token reduction. Although METR shows improvements in selecting less significant tokens to prune, it heavily relies on the presence of a CLS token and does not directly extend its benefits to non-classification tasks without CLS tokens. In contrast, our method can be applied without a CLS token and is applicable to various vision tasks.
>
> We will update our main paper to include a detailed discussion and comparison with both works.
>
> > W2. However, I do not believe using intermediate features directly is a drawback. The primary motivation of ToMe is to merge tokens with similar intermediate features. For example, if the similarity of two intermediate tokens is 1, merging these tokens is lossless, and additional embeddings are unnecessary in such scenarios. Therefore, the authors should provide more intuitive examples or an in-depth analysis beyond experimental comparisons to demonstrate the necessity of introducing additional embeddings.
>
> &rarr; In scenarios where token similarity is perfect (similarity equals 1), as the reviewer exemplified, merging based on intermediate features can indeed be lossless and sufficient. However, in practice, merging even highly similar tokens mostly leads to some degree of information loss, which becomes particularly significant when substantial token reduction is required. Thus, in practice, it is crucial for ViTs to preserve important information for tasks while merging tokens in areas that are less crucial for the task at hand. Note that this is a key heuristic also used in prior token pruning methods.
>
> The main limitation of relying solely on intermediate features for merging is that these features are optimized primarily for encoding, not for identifying less important regions. This can lead to indiscriminate token merging that sacrifices important details in more critical regions. In our method, decoupled embeddings encourage the reflection of the importance of information, guiding the merging process to occur predominantly in less important regions, as demonstrated in our appendix visualizations. These visual comparisons between traditional intermediate feature-based merging (ToMe) and our decoupled embedding-based approach (DTEM) clearly show that while intermediate features tend to merge tokens uniformly across background and object regions, our decoupled embeddings favor merging in the background, thus preserving valuable object information in the foreground.
>
> To further validate our intuition, we investigated whether the decoupled embeddings used for merging diverge from the intermediate features as learning progresses. We monitored changes in the Kendall rank correlation between token similarities derived from two different features: self-attention keys (as in ToMe) and decoupled embeddings. The results, shown in (Table S.4), indicate a decreasing correlation as learning progresses, suggesting that the decoupled embeddings seek a different measure of similarity for merging. This supports our intuition that the importance of token merging and token similarity with the intermediate features are not always aligned.
>
> Moreover, this intuition suggests that our method will be particularly effective under high reduction rates where information loss is more likely. This is aligned with our experimental results, indicating that our method becomes more effective with substantial token reduction.

---

> > ### Comment · Reviewer_wYAw · 2024-08-13
> >
> > Thank you for your response. I will maintain my positive rating. I have one question: You mentioned that the learnable metric guides the merging process in less important regions. DiffRate also sorts tokens by attention score, merging only the unimportant ones. How does the learnable representation compare to popular metrics like attention score in targeting less important regions? More comparison and discussion on this would be beneficial.

---

> > > ### Author Response · Authors · 2024-08-13
> > > **Thanks for the reply!**
> > >
> > > Thank you for your comments and the opportunity to clarify.
> > >
> > > Regarding your question about the comparison of our method to DiffRate, especially in the context of targeting less important regions for merging:
> > >
> > > &rarr; As the reviewer noted, methods like DiffRate sort tokens based on their importance, such as attention scores, and then reduce these (by merging or pruning) the relatively unimportant tokens. Consequently, DiffRate reduces only the less important tokens and minimizes information loss by ensuring that these tokens are merged with their most similar counterparts.
> > >
> > > In comparison, the uniqueness of our method lies in the use of the learnable decoupled embedding that can simultaneously consider both token redundancy and importance. This joint consideration is conducted through the single metric of similarity, enabling our method to not only focus on importance from a task perspective but also to assess the potential information loss, when selecting tokens to reduce. As a result, our approach may merge important tokens if such merging results in minimal information loss, or avoid merging less important tokens if no similar tokens are available, potentially leading to greater loss.
> > >
> > > Consider an example of two identical tokens in an important region. While approaches like DiffRate will avoid merging these due to their importance, our method will identify such pairing as an opportunity for lossless reduction, thus will merge them to decrease the number of tokens without losing information.
> > >
> > > We acknowledge that similar comparisons can be extended to related methods that use importance metrics and token merging, such as TPS [31]. We agree with the reviewer's comment that more comprehensive comparisons and discussions from these perspectives will further strengthen our paper. We will include more detailed discussions and comparisons in the revised version.
> > >
> > > If you have any further questions or remaining concerns, please let us know so we can address them.

---

### Official Review · Reviewer_FV5y · 2024-07-15

**Soundness:** 2
**Presentation:** 3
**Contribution:** 2
**Rating:** 4
**Confidence:** 4

**Summary:**

This paper works on the topic of visual token merging to improve the efficiency of ViT. Specifically, this work introduces decoupled token embedding for merging (DETM) which learns decoupled embedding via an additional module. Bu introducing the soft grouping and soft merging scheme, the proposed method is differentiable and thus,  parameters of the decoupled embedding module can be updated by gradient.

**Strengths:**

1. The paper is well written, which clearly demonstrates its motivation, methodology, experiment setup and the final results. Each part is easy to follow and the provided details should be enough for the re-implementation. Including the source code is a plus for re-implementation and understanding more details of the proposed method.

2. The token merging is a challenging topic in the community, which is still under exploration. The expensive computational cost of the transformer-based architecture has become the bottleneck and prevent it from further scaling up.  So, it is very important to invest effort in this topic to improve the throughput and efficiency of the ViT while maintaining/improving the performance.

3. This paper evaluate the effectiveness of the proposed method in various application direction, which is great. As is mentioned in the Line 68, these applications requiring a different level of granularity in representation.

**Weaknesses:**

1. It is argued that the similarity based token merging may not be optimal. This work introduces an additional embedding module to tackle this issue. So the decoupled visual embedding will be used for token merging only. It is concerned that if the capacity of proposed decoupled embedding module is able to learn different aspects compared to its input -- the original visual features. If so, how to regulate the decoupled feature is still representative of the original feature.

2. The original motivation of this work is to reduce the number of tokens and thus improve the efficiency. However, the proposed decoupled embedding module and iterative soft grouping would increase computational cost. There are some genera ablation study between the final performance and throughput/GFlLOPs. It is wondered how this is related to the hyperparameters in the above design.

3. Compared to previous works, the improvement of the proposed method is marginal, which generates the question that if it is worth to follow the design mentioned in this paper.

**Questions:**

Pure similarity based token merging may not be favorable. Besides, information will be lost with each merging. Is that possible to include additional cue to localize more important area and do token merging based on that?

The evaluation in this work is done in the image domain with standard ViT and configuration. In this case, the number of redundant token will not too much. It is also known that smaller patch size or larger input resolution may lead to better performance but suffer from the high computational cost. How about evaluate the token merging method in these scenarios?

**Limitations:**

mentioned in the weakness.

---

> ### Author Rebuttal · Authors · 2024-08-07
>
> > W1. It is concerned that if the capacity of the proposed decoupled embedding module is able to learn different aspects compared to its input--the original visual features. If so, how to regulate the decoupled feature is still representative of the original feature.
>
> &rarr; We clarify the main purpose and implementation of the decoupled embedding module.
> The module is specifically introduced to extract tailored embeddings suitable for merging from the original visual (intermediate) features. Thus, it takes the intermediate features as input and intended to output embedding vectors that are focusing on aspects beneficial for token merging. The module is trained via soft grouping and merging to emphasize merging advantageous aspects while relying on the original features.
>
> Our experimental results demonstrate that the decoupled embedding module, trained through soft grouping and merging, significantly alters the merging policy. Specifically, our method achieves improved performance and computational cost trade-offs. Moreover, the distinct token merging patterns are observed in our appendix visualizations, which show the difference between merging patterns using intermediate features and those merged using the decoupled features.
>
> We also note that the decoupled embeddings are utilized solely for the purpose of merging and are refined through a process of soft token merging. During such training, the remaining ViT parameters are not updated, ensuring that the original features are preserved by the learning of the decoupled embeddings.
>
> If there has been any misunderstanding of your concerns, we are willing to provide further clarification during the discussion period.
>
> > W2-1. The proposed decoupled embedding module and iterative soft grouping would increase computational cost.
>
> &rarr; We would like to clarify that our primary goal is to enhance computational efficiency during inference, or "test-time," which is crucial for real-world applications.
>
> Indeed, the soft grouping and merging operations incur additional computational overhead, but they are applied only during the training phase. In inference, despite the computation in the additional embedding module, our method offers an improved performance/computation cost trade-off (Figure 2).
>
> Moreover, our experimental results, as shown in Figure 5b, demonstrate that the decoupled embedding module and associated training processes are able to converge rapidly. This mitigates the potential increase in computational costs during the training phase. We also note that, in the end-to-end training scenario, we further optimize computational costs by updating the embedding module parameters less frequently, as detailed in lines 199-208 of the main paper.
>
> Thus, our method demonstrates a minimal increase in computational costs during training while significantly enhancing efficiency during inference.
>
>
> > W2-2. Impact of hyperparameters to our method.
>
> &rarr; To address the reviewer's concerns, we report more analysis related to hyperparameters, regarding (1) the number of steps in soft-grouping and (2) temperature scaling. We list the results in (Table S.1) and (Table S.2) of rebuttal pdf, respectively.
>
> For the number of steps (equals to the reduction rate in training) r in soft-grouping, we observe that the decoupled embedding module, when trained with a high reduction rate r, generalizes well to lower rates. Therefore, it is generally sufficient to set the number of steps to the maximum number of tokens we want to reduce during inference.
>
> For temperature scaling, we observed that values within the range of 0.1 to 0.3, tested with increments (0.05, 0.1, 0.2, 0.3, 0.5, 1.0), consistently provide gains with an accuracy difference of 0.1%.
>
> > W3. The improvement of the proposed method seems marginal.
>
> &rarr; While the improvements may seem modest, we believe they are reasonable and significant considering the differences from previous methods (as shown in Table 5).
>
> Moreover, our method introduces flexibility in training (modular and end-to-end) and achieves significant improvements compared to previous methods when trained modularly (Table 1). It even sometimes achieves performance comparable to that of end-to-end trained models (see Table 2).
>
> Additionally, unlike many existing methods that require separate models for different reduction rates, our approach works effectively across various rates with a single model, making deployment simpler and more cost-effective.
>
>
> > Q1. Is it possible to include additional cues to localize more important areas and do token merging based on that?
>
> &rarr; We interpret this question as inquiring whether it is possible to consider the importance of tokens (or areas) when conducting token merging. We argue that our method inherently incorporates this aspect into its merging policy. In our method, the regions where tokens are merged (as shown in the appendix visualization) demonstrate that merging predominantly occurs in less important background areas. This pattern emerges because our decoupled embedding is specifically trained to recognize and prioritize tokens of lesser importance for classification. This indicates that our method learns to define similarity based on importance, favoring merging in less important regions, thereby preserving essential information in areas of greater significance.
>
> > Q2. How about evaluating the token merging method in smaller patch size or larger input resolution settings?
>
> &rarr; Following the reviewer's comment, we conducted experiments with a larger number of tokens on image classification (ViT-small with resolution 384x384), which corresponds to smaller patches or higher input resolutions.
>
> The results (Table S.3 in rebuttal pdf) show that our method can adapt to settings with an increased number of tokens, achieving performance gains. We also note that the segmentation tasks were conducted at a resolution of 512x512.

---

### Official Review · Reviewer_PcYu · 2024-07-15

**Soundness:** 2
**Presentation:** 3
**Contribution:** 2
**Rating:** 6
**Confidence:** 3

**Summary:**

In this paper, decoupled token embedding for merging (DTEM) is proposed for more efficient and effective token merging.
It employs a lightweight embedding module to obtain a feature vector which is solely used for token merging process. To train this embedding module, DTEM uses the relaxed merging method based on continuous matching scores.
Extensive experimental results on image classification, image captioning, and segmentation show that the proposed algorithm outperforms the conventional token merging algorithms.

**Strengths:**

1. The paper is well-written and easy to understand.
2. The proposed algorithm is simple but technically sounds.
3. The proposed algorithm achieves the better performances than other token merging methods on various datasets.

**Weaknesses:**

1. Why [18] is not compared in Table 1?
2. It would be better if there is the analysis on alternative design choices for soft grouping (eq 6-7) and soft merging (eq 9-10).
3. In Figure 3, the results within the limited reduction range (31-41% or 31-49%) are reported. More results with diverse reduction rates would be helpful to validate the effectiveness of the proposed algorithm.
4. The discussion on the experimental results in Section 4.4 is minimal. More explanation would be helpful for readers.

Minor:
5. Figure 3 and 4 are Tables, as referred in L284 and L302.

**Questions:**

In overall, I could not find the critical flaws in this work. Even though the proposed algorithm is simple, but it technically sounds to me. Also, it shows good performances over various computer vision tasks. For my concerns, please see the weakness section.

**Limitations:**

Yes, in Section A.5.

---

> ### Author Rebuttal · Authors · 2024-08-07
>
> > W1. Why is [BAT, 18] not compared in Table 1?
>
> &rarr; Table 1 showcases the results for methods applied to a pretrained, frozen ViT with modular training, whereas BAT was originally proposed and evaluated in an end-to-end training setting without results from modular training. While it seems possible to apply BAT to a frozen ViT, the lack of a publicly available implementation for BAT precludes us from conducting meaningful adaptations or comparisons under this condition (only training logs are publicly available).
>
> > W2. It would be better if there is an analysis on alternative design choices for soft grouping (eq 6-7) and soft merging (eq 9-10).
>
> &rarr; Following the reviewer's comment, we conducted further analysis for soft grouping and merging.
>
> For the analysis on soft grouping, we compared several approaches: (1) integrating a Gumbel softmax with the top-1 operation from ToMe to enable differentiation, (2) applying DynamicViT to a frozen ViT setting (pruning), and (3) our method, which uses a modified relaxed top-k operation, in (Table S.5). The results indicate that our proposed design for soft grouping performs the best.
>
> Next, we experimented with hard merging, using discretization and actual discarding of tokens. We observed that this led to divergence in training, confirming that soft merging is essential for our method.
>
>
> > W3. In Figure 3, the results within the limited reduction range (31-41% or 31-49%) are reported.
>
> &rarr; Following the reviewer’s suggestion, we report Figure 3 results across a broader reduction range, in (Table S.7). The result shows that our method is particularly effective in challenging, more resource-constrained settings with higher reduction rates.
>
> > W4. The discussion on the experimental results in Section 4.4 is minimal.
>
> &rarr; We acknowledge the need for a more detailed explanation in Section 4.4, particularly concerning the importance of decoupled embedding (Table 4) and module design (Figure 6).
>
> To be more specific, in Table 4, we successively add components: (1) soft token merging and (2) decoupled embedding module to ToMe. When only (1) soft-token merging is applied, the gradient from soft-grouping and merging is directly passed through the intermediate features of ViT. The results show that not only the gradients from the merging policy (with soft-grouping and merging) but also the decoupled embedding module, detached from the intermediate features of ViT, are crucial.
>
> In Figure 6, we experimented with an MLP embedding module and demonstrated that (1) an affine transformation is sufficient while an embedding dimension of 64 provides the best trade-off between computation cost and performance.
>
> In the revised version of the manuscript, we will make the description more comprehensive and clear.
>
>
> > W5 (Minor). Figure 3 and 4 are Tables, as referred in L284 and L302.
>
> &rarr; We thank you for pointing it out. We will correct it in the revised version.

---

> > ### Comment · Reviewer_PcYu · 2024-08-14
> >
> > I appreciate the detailed responses, which have addressed most of my concerns. I also have read the reviews from other reviewers, but I still believe that the proposed algorithm has some meaningful contribution to the community. So I decide to keep my original rating.

---

> > > ### Author Response · Authors · 2024-08-14
> > > **Thanks for the reply!**
> > >
> > > Thank you for your reply. We are happy to hear that we have addressed most of your concerns. If you have any further questions or concerns, please do not hesitate to let us know. Although the discussion period is nearing its end, we will do our best to address any remaining issues.

---

### Author Rebuttal · Authors · 2024-08-07

We thank the reviewers for their time and effort in providing constructive reviews. We appreciate the encouraging remarks about the paper's novelty (oGzP), technical soundness (PcYu), promising experimental results (oGzP), and applicability (oGzP, wYAw, FV5y). We are happy to respond to the weaknesses (W) and questions (Q) in the comments and hope that our responses address your concerns. Due to the text limit, we have included all rebuttal experimental results (Tables S.1 to S.7) in the rebuttal PDF.

---

### Decision · Program_Chairs · 2024-09-25

**Decision:**

Accept (poster)

**Comment:**

This paper proposed a decoupling method to enable continuous relaxation of token merging. The novelty is well-recognized by the reviewers. The authors demonstrate the effectiveness of the proposed method via extensive experiments. Three reviewers provided positive feedback, except for the reviewer FV5y, who has major concerns about the side effects of the parameter reduction. The authors carefully addressed the raised issues. Unfortunately, the reviewer  FV5y didn't provide further feedback. The Meta-review agrees with the clarity presented in the author's rebuttal.